# *PmbHLH58* from *Pinus massoniana* Improves Drought Tolerance by Reducing Stomatal Aperture and Inducing ABA Receptor Family Genes in Transgenic Poplar Plants

**DOI:** 10.3390/ijms26010277

**Published:** 2024-12-31

**Authors:** Jinfeng Zhang, Manqing Peng, Peizhen Chen, Sheng Yao, Yuan He, Dengbao Wang, Romaric Hippolyte Agassin, Kongshu Ji

**Affiliations:** State Key Laboratory of Tree Genetics and Breeding, Key Laboratory of Forestry Genetics & Biotechnology of Ministry of Education, Co-Innovation Center for Sustainable Forestry in Southern China, Nanjing Forestry University, Nanjing 210037, China; zhangjinfeng@njfu.edu.cn (J.Z.); pmq@njfu.edu.cn (M.P.); pei_jane@njfu.edu.cn (P.C.); shengyao@njfu.edu.cn (S.Y.); heyuan@njfu.edu.cn (Y.H.); dbw@njfu.edu.cn (D.W.); asnromaric@njfu.edu.cn (R.H.A.)

**Keywords:** *Pinus massoniana* Lamb., bHLH transcription factors, *PmbHLH58*, drought tolerance, PmERF71

## Abstract

The basic helix–loop–helix (bHLH) family members are involved in plant growth and development, physiological metabolism, and various stress response processes. *Pinus massoniana* is a major turpentine-producing and wood-producing tree in seasonally dry areas of southern China. Its economic and ecological values are well known. The forestry industry holds it in exceptionally high regard. Drought severely limits the growth and productivity of *P. massoniana*, and the functional role of *PmbHLH58* in drought stress is not clear. Therefore, *PmbHLH58* was cloned from *P. massoniana* and its bioinformation was analyzed. Subcellular mapping of the gene was performed. The biological function of *PmbHLH58* overexpression in *Populus davidiana* × *P. bolleana* was studied. The results show that the drought tolerance of *PmbHLH58*-overexpressed poplar was significantly improved, which may be due to the increase in water use efficiency and reactive oxygen species (ROS) accumulation under drought stress. In an ethylene-responsive manner, PmERF71 interacted with the PmbHLH58 protein, which was found by yeast two-hybridization. We further demonstrated that the drought-induced *PmbHLH58* transcription factor increased the expression of key enzyme genes in ABA receptor family genes in *PmbHLH58*-overexpressing poplar lines (OE). These findings provide new insights into transcriptional regulation mechanisms related to drought stress and will promote the progression of the genetic improvement and plantation development of *P. massonsiana*.

## 1. Introduction

Drought is one of the most important abiotic factors that hinder plant growth. As the global climate changes, temperatures and droughts will continue to increase (Global Drought Report, January 2024). Over the past decade, global crop yield losses due to drought have totaled about USD 30 billion (Food and Agriculture Organization of the United Nations (FAO), 13 October 2023). Therefore, we need to take corresponding measures to improve stress resistance in plants. Plant drought tolerance can be enhanced by adjusting stomatal movement, osmotic substance content, root structure, and antioxidant ability. Under drought stress, plants reduce water loss and improve their water use efficiency mainly through stomatal closure, thus improving drought resistance [1]. Stomatal closure is largely regulated by cellular signaling mediated by ABA, ROS, and Ca^2+^ [2]. The roots produce a large amount of ABA, which is transported to the leaves through the xylem and induces stomatal closure under drought stress [3]. Under drought conditions, ROS are important second messengers in stomatal movement, which can lead to rapid stomatal closure at a certain concentration [4,5]. ABA and ROS are closely related. ABA can induce ROS production in protective cells [6], leading to stomatal closure, further reducing leaf transpiration rates and water loss. As a second messenger, Ca^2+^ ions are usually involved in regulating the stomatal movement, antioxidant enzymes, and aquaporin activities of plants and improving water use efficiency to enhance drought resistance [7].

*Pinus massoniana* Lamb. is a vital turpentine-producing and wood-producing tree species in China, which is widely used in afforestation, industrial timber, and resin production. It is mainly distributed in the vast areas south of the Qinling and Huaihe River [8]. Seasonal drought is threatening the growth and development of *P. massoniana* due to the uneven distribution of annual rainfall. A method to improve the drought tolerance of *P. massoniana* is urgently needed. Scholars at home and abroad have made significant progress in drought tolerance mechanism research and related gene cloning and expression regulation. Cai Q. and Fox A.R. studied the transcriptomic response to drought stress of *P. massoniana* [9,10], and the genes of *PmWRKY*, *PmNAC*, *PmAOX*, *PmSnRK2s*, and *PmERF1* associated with abiotic stress resistance were cloned and their functions were analyzed [11,12,13,14,15]. These studies were helpful for research on the abiotic stress resistance of *P. massoniana.*

The response of plants to drought is complicated by diverse proteins and metabolites that form a complex regulatory network in plant defense [16]. bHLH (basic helix–loop–helix) is widespread in eukaryotes, and its family is the second-largest transcription factor (TF) family after the MYB family in plants [17]. bHLHs have a highly conserved helix–loop–helix domain with two functionally distinct regions (N-terminal and C-terminal) [18]. The N-terminal binds E-box(CANNTG) or G-box(CACGTG). The C-terminal is involved in the binding of homodimers or heterodimers, thereby regulating the expression of downstream genes [19]. bHLHs are essential in regulating plant growth and development and resisting abiotic stress (such as drought, high salt, and low temperatures) [20]. There have been a great deal of reports on the involvement of bHLH in plant responses to stress. The overexpression of *FvICE1* improved the drought tolerance of *Fragaria vesca* [21]. *AtbHLH122*-overexpressing plants had increased resistance to abiotic stress [22]. The overexpression of ZmbHLH124T-ORG in maize and rice improved drought tolerance by up-regulating the expression of drought-responsive genes [23]. Under drought stress and ABA treatment, PebHLH35 expression was up-regulated in *Populus euphratica*, and overexpression of *PebHLH35* in *Arabidopsis thaliana* improved resistance to drought by regulating stomatal density and stomatal size [24]. Besides bHLHs, there are many other TFs, such as NAC, AP2/ERR, WRKY, MYB, and b-ZIP, involved in stress responses [25,26,27,28]. When MbWRKY3 was introduced into tobacco, it improved the drought tolerance of the transgenic plants [29]. The overexpression of *VlbZIP30* can reduce the water loss rate, maintain an effective photosynthetic rate, and improve drought tolerance under drought conditions [30].

The aim of this study was to expand the regulatory role of bHLH in drought tolerance; we selected highly expressed bHLH genes from transcriptome data of drought stress for cloning. At present, the genetic transformation system of *P. massoniana* has not been established. Therefore, the gene was transferred into *Populus davidiana* × *P. bolleana*, and the transgenic plants, as well as wild-type plants, were subjected to drought treatment. The physiological indexes and photosynthetic characteristics were analyzed during the drought treatment to determine if the transgenic plants had improved drought tolerance compared to the wild-type plants. The acquisition of forest products is closely related to the improvement of the stress resistance of cultivated tree species. The functional analysis of the *PmbHLH58* gene can lay the foundation on the response mechanism of *P. massoniana* to drought stress, and furnish reference for its breeding for resistance gene engineering.

## 2. Results

### 2.1. Expression Pattern Analysis of PmbHLH58

The *PmbHLH58* (c129441.graph_c1) gene expression level was significantly up-regulated from the drought stress transcript data (SRA accession: PRJNA595650). In the functional annotation, *PmbHLH58* was involved in plant hormone signal transduction and circadian rhythm, suggesting that bHLH may be involved in the response of *P. massoniana* to drought stress. NCBI software analyzed the gene as *PmbHLH58*, which has an ORF of 1854 bp and encodes 617 amino acids and 1 stop codon. Heim et al. [31] divided the bHLH transcription factors in *A. thaliana* into 12 subgroups based on sequence similarity, and the evolutionary analysis showed that PmbHLH58 belonged to family VIIb, and that it is on the same evolutionary branch as the reported PIF3, PIF5, PIF4, and PIF6 subfamilies (Figure 1A). To further explore the function of *PmbHLH58*, the homologous gene series of six species, including *P. massoniana*, *Taxus chinensis*, and *Cryptomeria japonica*, were used to construct the evolutionary tree (Figure 1B). The PmbHLH58 protein showed high sequence similarity with *T. chinensis* and is relatively close to *C. japonica*. According to NCBI Conserved Domains software analysis, the protein has a bHLH_*AtPIF*_like conserved domain, and the domain belongs to the bHLH_SF supergene family (Figure 1C). ExPasy ProtParam analysis of the PmbHLH58 protein in *P. massonsiana* showed that its molecular formula is C_5534_H_9216_N_1854_O_2277_S_435_, its molecular weight is 152,103.20 Da, and its theoretical iso-electric point (PI) is 4.93. The hydrophilic average (GRAVY) is 0.861, which means it is a hydrophobic protein.

To determine the subcellular localization of *PmbHLH58*, transient expression vectors of the protein (GFP) (35S::PmbHLH58-GFP and 35S::-GFP) were transferred into *N. benthamiana* leaves. Under a confocal laser scanning microscope, the 35S::PmbHLH58-GFP fusion protein was localized in the nucleus, while the 35S::-GFP fusion protein had no specific localization (Figure 1D). These results confirm that the PmbHLH58 protein is nuclear-localized.

We investigated the expression profiles of *PmbHLH58* in various tissues of *P. massoniana* (young needles, old needles, young stems, old stems, roots, xylems, and primary xylems) by RT-qPCR. *PmbHLH58* transcript needles were higher in young needles than in young stems and primary xylems (Figure 1E). In order to study the response of *PmbHLH58* to drought stress, the expression level of *PmbHLH58* using qRT-PCR under ABA and SA was tested. After ABA treatment, the *PmbHLH58* transcript level was gradually increased, reaching a peaked at 6 h, growth about 2.5 times, and decreased thereafter (Figure 1F). The expression of *PmbHLH58* was 10 times higher than that in the WT after 3 h with 5 mM SA (Figure 1G). These results suggest that *PmbHLH58* was mainly expressed in young needles and is up-regulated under ABA and SA treatment.

### 2.2. Heterologous Expression of PmbHLH58 in Populus davidiana × P. bollean

In order to study the role of *PmbHLH58* in plants under drought stress, the leaf disk method was adopted to transform the pSuper:: *PmbHLH58* carrier into woody wild-type (WT) *Populus davidiana* × *P. bolleana* (Appendix A). The RT-PCR and qRT-PCR results showed that the *PmbHLH58* was integrated into five independent transgenic plant lines and was highly expressed in L2 and L6 (Appendix A). Therefore, we selected these lines as experimental materials for subsequent analyses.

### 2.3. PmbHLH58 Promotes ABA-Induced Stomatal Closure

H_2_O_2_ and MDA content are the two main indexes of ROS level and oxidative damage [32]. ABA-induced stomatal closure was accompanied by H_2_O_2_ production and thus participates in the closing process of guard cells [33,34]. To confirm whether *PmbHLH58* is involved in ABA-induced ROS signal transduction, instantaneous H_2_O_2_ and MDA contents in the WT, 35S::-GFP, and OE line plants were measured, respectively. Under ABA treatment, the H_2_O_2_ and MDA contents of the OE line, 20 35S::-GFP, and WT plants increased to a certain extent. However, the increase of H_2_O_2_ and MDA contents in the OE strain was significantly lower than that in the WT strain (Figure 2A,B). The results showed that the OE lines could clear ROS and improve drought tolerance.

H_2_O_2_ is a type of reactive ROS which is an essential second messenger in stomatal movement and can cause rapid stomatal closure at certain concentrations [5]. To investigate whether PmbHLH58 promoted ABA-induced stomatal closure, leaf stomatal movement of the wild type, the 35S::-GFP and OE lines after ABA treatment was observed resepectively by scanning electron microscopy. After 1 h and 2 h treatment with 0.5 mM ABA, we observed that the stomatal closure speed of transgenic plants was faster than that of wild-type plants (Figure 2C), and the stomatal pore size decreased significantly (Figure 2D). PmbHLH58 promotes ABA-induced stomatal closure.

### 2.4. Under Short-Term and Long-Term Drought Stress OE Lines Showed Improved Drought Tolerance

In order to discuss the difference between the WT, 35S::-GFP, and OE lines under drought stress, a drought treatment was started by withholding watering for 7 days. At 10:00 am daily, the net photosynthetic rate (*Pn*), transpiration rate (*Tr*), and stomatal conductance (*Gs*) of the leaves were measured. As predicted, the leaves of WT plants were severely withered after 7 days of drought stress, while those of the OE lines remained swollen (Figure 3A). Under drought treatment, the photosynthetic rate of WT plants decreased markedly. The photosynthetic activity of the plants was almost zero on the 7th day of drought treatment. In contrast, the *Pn* of the OE lines reduced significantly from the 1st to the 4th day, then gradually slowed down and remained at a certain level until the 7th day (Figure 3B). These results indicate that under drought conditions OE plants can still produce the energy for photosynthesis. The Gs and transpiration rate in the WT, 35S::-GFP, and OE lines showed a consistent downward trend, but the decline rate in the OE lines was slower than that in the WT plants under drought stress (Figure 3C,D). In addition, under drought conditions, the RWC of OE leaves was higher than that of the WT plants (Figure 3E). In conclusion, the tolerance of OE was higher than that of WT plants under short-term drought stress.

To further explore the role of *PmbHLH58* in long-term drought stress, we carried out long-term drought treatment for the WT, 35S::-GFP, and OE for 60 days, and maintained a soil relative water content of 70% (control) and 30% (drought). After 60 days, the leaves of OE remained swollen and had better growth status than those of WT (Figure 4A). In addition, the RWC of the leaves of OE was higher than in WT plants, indicating that OE had stronger drought tolerance (Figure 4B). Photosynthesis analysis showed that the photosynthetic rate of the OE lines was not significantly different from that of WT plants in the control (70% soil RWC). However, OE had a higher photosynthetic rate than WT plants under drought conditions (30% soil RWC) (Figure 4C).

By measuring the content of chlorophyll a and chlorophyll b, we found that under 30% RWC conditions, the contents of chlorophyll a and chlorophyll b in the OE lines were significantly higher than those in 35S::-GFP and WT plants, while there was no significant difference among WT, 35S::-GFP and OE plants under 70% RWC conditions (Figure 5A,B). These results showed that OE’s light energy absorption capacity is better than that of 35S::-GFP and WT plants under long-term drought, so OE could maintain higher photosynthetic rates. To investigate the growth difference between the WT, 35S::-GFP, and OE lines, we measured the plants’ stem elongation rates and biomass under drought conditions. Under 70% RWC, the stem elongation rate of OE was not significantly different from that of 35S::-GFP and WT plants, but under 30% RWC conditions, the stem elongation of OE was significantly higher than that of 35S::-GFP and WT plants (Figure 5C). The biomass accumulation of OE was significantly higher than that of WT plants, in accordance with the photosynthetic rate, which was higher by 51.23% and 47.32%, respectively, after long-term drought treatment. However, under well-watered conditions, there was no significant difference between the OE and WT plants (Figure 5D). Therefore, under long-term drought conditions, the overexpression of *PmbHLH58* can improve the drought tolerance of plants and is conducive to plant growth.

### 2.5. The PmbHLH58 Protein Interacts with PmERF71

Homologous genes in the same species have very similar functions. *PmERF71* homologous gene *PmERF1* may plays an active role in drought stress tolerance by improving WUE via lower transpiration and increasing the expression of ABA signaling-related genes [15]. We screened from yeast two-hybrid experiments that the *PmbHLH58* gene could interact with ethylene response transcription factor, E3 ubiquitin protein, superoxide dismutase, proline-rich receptor-like protein kinase, and severe drought stress response protein to regulate the drought stress of *P. massoniana* (Table 1).

To investigate whether the PmbHLH58 protein interacts with the critical drought-related enzyme PmERF71 and ubiquitin–protein ligase PmPUB22, respectively, we constructed the pGADT7-PmERF71 and pGADT7-PmPUB22 vectors and conducted point-to-point Y2H verification with pGADT7-PmbHLH58. The results show that the positive control strains could grow normally on DDO, TDO, and QDO; negative control strains grew on DDO but did not grow on TDO and QDO. In the experimental group, pGBKT7-PmbHLH58 *+* pGADT7-PmERF71 grew on both TDO and QDO, while the growth of pGBKT7-PmbHLH58 *+* pGADT7-PUB22 was the same as that of the negative control (Appendix A). This indicated that pGBKT7-PmbHLH58 + pGADT7-PmERF71 interacted with the pGADT7-PmERF71 vector but not with pGADT7-PUB22. The protein was diluted and knotted on DDO and QDO. The results of point-plate verification are consistent with those of rotation verification (Figure 6).

To verify the authenticity of the interaction, we investigated the PmbHLH58-PmERF71 interaction in vivo. BiFC tests were executed in *N. benthamiana* leaves by laser scanning confocal microscope. According to the fluorescence results, the YFPN-PmbHLH58 + PmERF71-YFPC experimental group and positive control groups showed strong fluorescence, while the other experimental and negative control groups showed no fluorescence (Figure 7).

### 2.6. Analysis of Key Enzyme Genes in ABA Hormone Signaling Pathway

ABA accumulation in plant cells can activate downstream signaling components [35]. We speculate that *PmbHLH58* may be involved in the ABA regulatory pathway through interaction with *PmERF71*. To further confirm this hypothesis, we used qRT-PCR analysis to analyze the expression profiles of the abscisic acid receptor family genes (PYL5, PYL6, SRK2, ARF5, ERF71, MYC2). The results indicated these genes were up-regulated in OE (Figure 8). Therefore, the expression trend of the key enzyme genes in the hormone signal transduction pathway of OE was consistent with the scanning phenotype of the electron microscope, thus improving plant drought tolerance.

## 3. Discussion

Drought seriously affects the growth and development of plants. The bHLH family is involved in various biological and abiotic stress responses [20]. Researchers have unraveled how plants cope with drought stress by producing the phytohormone ABA, reprogramming gene expression, closing stomata, and making osmotic adjustments, eventually leading to adaptive growth and development [36]. However, the function of the bHLH family gene in *P. massoniana* has not been studied. In this study, we identified a new drought-resistance gene PmbHLH58 and predicted its function, which provided a theoretical basis for the study of the drought tolerance of *P. massoniana*.

The bHLH gene has been extensively studied in many plants, such as *MdSAT1*, Trifoliate orange, and *Quercus mongolica* [26,37,38]. But it is rarely reported in *P. massoniana*. In this study, the relatively high expression of the *PmbHLH58* gene was screened from drought stress transcriptome data. Using *P. massoniana* cDNA as a template, the complete CDS sequence of *PmbHLH58* was amplified by RT-PCR. PmbHLH58 is 1854 bp long and encodes 617 amino acids and 1 stop codon. Since genes of the same gene family are annotated to specific groups in different species, they also have different functions [39]. Therefore, it is very important to classify the *PmbHLH58* gene in detail and interpret its function according to the model plant *Arabidopsis*. Phylogenetic tree analysis showed that *PmbHLH58* belongs to the subfamily VIIb, which is similar to the reported subfamilies PIF3, PIF5, PIF4, and PIF6 (Figure 1A). We found that ABA and SA treatments induced *PmbHLH58* expression (Figure 1G,F). In addition, *PmbHLH58* was mainly expressed in young needles (Figure 1E). These results indicate that *PmbHLH58* may be involved in the response process of *P. massoniana* to drought stress.

ROS is critical in regulating stomatal closure and responses to adverse environmental conditions such as drought [4,40,41]. When plants are stimulated by stress, intracellular ROS accumulate in large quantities, causing stomatal closure [42]. *AhbHLH112* can improve ROS clearance by regulating POD-mediated H_2_O_2_ homeostasis, and may participate in the ABA-dependent stress response pathway. Overexpressed plants have a stronger tolerance to drought stress [43]. This study showed that the H_2_O_2_ content and MDA content in OE, 35S::-GFP, and WT plants increased after drought stress. However, the increase in H_2_O_2_ content in OE was significantly lower than that in WT and 35S::-GFP plants, and MDA content was significantly lower than that in WT plants (Figure 2A). In this study, the leaf stomatal pore size of OE was significantly smaller than that of WT and 35S::-GFP plants under drought stress (Figure 2C). Therefore, this study suggests that *PmbHLH58*-overexpressing plants show stronger drought tolerance.

Plants can reduce water demand and prevent water loss by reducing stomatal conductance under drought stress [44]. Studies have shown that the decrease in stomatal conductance was contrary to photosynthesis and biomass accumulation [45], while some studies have shown that the decrease in stomatal conductance was not related to photosynthesis but to photosynthetic enzyme activities [46]. In this study, the *Pn*, *Tr*, and *Gs* of OE, WT, and 35S::-GFP plants decreased with the increase in drought stress duration. On the 4th day of drought stress, the leaves of WT and 35S::-GFP plants showed a wilting state, and the *Pn*, *Tr*, and *Gs* were lower than those of OE. After 7 days of drought, the degree of leaf wilting of the OE lines was much lower than that of WT and 35S::-GFP plants, and OE could still grow through photosynthesis to a certain extent, which was also consistent with the phenotype observation (Figure 3A–C). Lower photosynthetic indices prevent excessive water loss, which in turn reduces photosynthesis and biomass accumulation [44]. There was no significant difference in the growth rate and biomass of each line under the non-stress conditions in the long-term drought test (Figure 5C,D). This may be caused by the high relative water content of the leaves of overexpressed strains (Figure 3D). The results showed that the overexpression of PmbHLH58 can improve drought tolerance by regulating stomatal opening and photo-cooperation.

Drought limits plant growth, development, quality, and yield [47]. The drought tolerance of plants is complex and it often involves the network regulation of multiple genes. bHLH plays an active role in plant response to drought stress. In the past, researchers have revealed how bHLH family genes respond to drought stress in plants by regulating related synthetic pathways of abscisic acid, stomatal closure, photosynthesis, etc., ultimately leading to adaptive growth and development [48]. In this study, we identified the drought tolerance gene *PmbHLH58*. The protein has a bHLH_AtPIF_like conserved domain and belongs to the bHLH_ supergene family (Figure 1C). Protein-to-protein interactions help to explore new functions of proteins [49]. To further understand the mechanism of PmbHLH58, we used PmbHLH58 as the basic protein and screened 28 proteins from the *P. massonsiana* yeast library. Various validation methods are used to check interactions between proteins, such as Y2H, BiFC, etc. [50]. In this study, the interaction between the PmbHLH58 and PmERF71 proteins was verified using the Y2H and BiFC techniques (Figure 7). We found that essential enzyme genes in the ABA hormone signaling pathway, such as *PYL5*, *PYL6*, *SRK2*, *ARF5*, *ERF71*, or *MYC2*, were up-regulated in the transgenic plants (Figure 8). The overexpression of PmbHLH58 may be involved in ABA-dependent stress response pathways to improve drought tolerance (Figure 9). In addition, the transgenic poplar obtained in this paper also has far-reaching practical value for the further breeding and popularization of new varieties of drought-resistant poplar.

## 4. Materials and Method

### 4.1. Plant Materials

Two-year-old *P. massoniana* seedlings with stable growth status were used for the experiments. Eight kinds of tissues, including root, young needle, old needle, young stem, old stem, xylem, and primary xylem, were collected and immediately frozen in liquid nitrogen. In addition, for ABA and salicylic acid (SA) treatment, aqueous solutions of 0.5 mM ABA and 1.0 mM SA were sprayed on the needles, respectively, and these were then covered with a clear plastic film [51]. Needles for each test were collected at 0, 3, 6, 12, and 24 h, respectively, and immediately frozen in liquid nitrogen and stored at −80 °C.

### 4.2. cDNA Cloning of PmbHLH58 from P. massoniana and Quantitative Real-Time PCR Analysis

PmbHLH58 was cloned from a cDNA library constructed with RNA from needle tissue. RNA prep Pure Plant Kit (Polysaccharides & polyphenolic-rich) (Tiagen Biotech, Beijing, China) was used to isolate total RNA from each sample. The NanoDrop 2000 instrument (Thermo Fisher Scientific, Waltham, MA, USA) was used to detect its concentration. The RNA was reverse transcribe into cDNA using the 1st-strand cDNA synthesis kit (11141, Yeasen Biotech, Shanghai, China). All primers used in these assays are listed in Appendix A.

Quantitative real-time PCR (RT-qPCR) was performed using the SYBR Green Real-time PCR Master Mix (QPK-201, Toyobo Bio-Technology, Shanghai, China) according to the manufacturer’s instructions. Each PCR mixture (10 µL) contained 1 µL of diluted cDNA (20× dilution), 5 µL of SYBR Green Real-time PCR Master Mix, 0.4 µL of each primer (10 µM), and 3.2 µL of ddH_2_O. The PCR program had six stages: (1) 95 °C for 60 s (pre-incubation); (2) 95 °C for 15 s; (3) 60 °C for 15 s; and (4) 72 °C for 10 s, repeated 40 times (amplification); (5) 95 °C for 0.5 s; and (6) 60 °C for 1 min (melt). TUA (α-tubulin) was used as the internal control [52]. Each biological replicate was examined three times. Gene expression levels were calculated using the 2^−∆∆Ct^ method described by Livak and Schmittgen [53]. The significance was determined by one-way analysis of variance using SPSS statistical software (IBM, New York, NY, USA, https://www.ibm.com/products/spss-statistics) (*p* < 0.05).

### 4.3. PmbHLH58 Protein Phylogenetic Tree Reconstruction and Domain Analysis

The *Arabidopsis* bHLH gene sequence of acids was accessed from the TAIR database (http://www.Arabidopsis.org/, accessed on 22 August 2023). The conserved domain of the PmbHLH58 protein was analyzed in the NCBI Conserved Domain database (http://www.ncbi.nlm.nih.gov/Structure/cdd/wrpsb.cgi, accessed on 22 August 2023). And the sequences were used to construct a phylogenetic tree by the neighbor-joining (NJ) method using a Poisson model in MEGAX software (Park, PA, USA) [54] (https://www.megasoftware.net/, accessed on 22 August 2023), and multiple sequence alignment was carried out with DNAMAN 8 software [55]. Then, we used iTOL software (https://itol.embl.de/, accessed on 22 August 2023) to edit the phylogenetic tree.

### 4.4. Subcellular Localization

The nuclear localization signal (NLS) of PmbHLH58 was predicted by Cell-PLoc 2.0 (http://www.csbio.sjtu.edu.cn/bioinf/Cell-PLoc-2/, accessed on 10 September 2023). To confirm the predicted result, we transformed the 35S:: PmbHLH58-GFP and helper plasmid Pjit166 into the tobacco leaves. The fluorescence signals were observed with an LSM 710 confocal microscope (Zeiss, Jena, Germany), which was also used to observe fluorescence signals.

### 4.5. Yeast Two-Hybrid Assay

The *PmbHLH58* sequence was cloned into vector pGBKT7 and the recombinant pGBKT7-PmbHLH58 plasmid was used as bait while using the *P. massoniana* yeast library plasmids as prey. The pGBKT7-PmbHLH58 construct and library plasmids were co-transformed into the Y2H Gold yeast strain following the manufacturer’s protocol (Takara, Kyoto, Japan). Subsequently, these yeast cells were cultured on SD-Trp-Leu (DDO), SD-Trp-Leu/His/-Ade (QDO), SD/-Leu/-Trp/-His (TDO), and SD/-Leu/-Trp/-His/X-α-Gal (TDO/X) plates.

### 4.6. Point-to-Point Verification

We selected the ethylene-responsive transcription factor ERF071 (XM_035048879.1) protein and the E3 ubiquitin–protein ligase PUB22 (XM_035061321.1) protein from the yeast library screening results to verify whether they interacted with PmbHLH58 by yeast two-hybrid experiment. The ORFs of the PmERF071 and PmPUB22 genes were inserted into the digested vector by recombinant technology.

### 4.7. Bimolecular Fluorescence Complementation (BiFC) Assays

The PmbHLH58 coding sequence was cloned into the pSPYNE vector to generate YFPN-PmbHLH58. The *PmERF71* coding sequence was inserted into the pSPYCE vector to generate PmERF71-YFPC. The construct was introduced into *Agrobacterium tumefaciens* strain GV3101 cells and then resuspended in infiltration buffer. The *A. tumefaciens* cells harboring YFPN-PmbHLH58 or YFPN were mixed with cells carrying PmERF71-YFPC for 2 h, and were then used to co-infiltrate *Nicotiana benthamiana* leaves by needleless syringes. Fluorescence was detected with an LSM 710 confocal microscope (Zeiss, Jena, Germany).

### 4.8. PmbHLH58 Overexpression in Populus davidiana × P. bolleana

To construct the *PmbHLH58* overexpression vector, the CDS sequence of PmbHLH58 was inserted into the pBI121 vector by two restriction endonuclease enzymes XbaI and SmaI. The recombinant plasmid was introduced into the *A. tumefacient* strain GV3101 by heat shock and transformed into the wild-type *Populus davidiana* × *P. bolleana* by the leaf disk method [56]. The genomic DNA of non-transgenic plants (WT) and transgenic poplar (OE) lines was extracted by the CTAB method [57]. The inverters were identified by PCR using the forward primers of the CAMV 35S promoter and the reverse primers for *PmbHLH58*, then *PmbHLH58*-overexpressing transgenic poplar lines (OE) were generated.

The transcriptional levels of transgenic poplar *PmbHLH58* were verified by RT-qPCR. Total RNA was extracted from the leaves of WT and OE plants, respectively. The first-strand cDNA synthesis was performed with the One-step gDNA removal and cDNA Synthesis Kit (AT311, TransGen Biotech, Beijing, China) according to the manufacturer’s instructions. The primers are listed in the Appendix A.

### 4.9. Drought Experiments

In the short-term drought experiment, 20 wild-type (WT), 20 35S::-GFP, and 40 OE plants growing in pots (140 cm width and 125 cm height) for 2 months were used as experimental materials. The plants were not watered for 7 days as drought stress. Soil relative water content (RWC), stomatal conductance (*Gs*), transpiration (*Tr*), and net CO_2_ assimilation (*Pn*) were measured daily under drought stress.

In the long-term drought experiment, 20 WT, 20 35S::-GFP, and 40 OE plants were planted in pots (140 cm width and 125 cm height) under soil moisture content conditions of 70% (no stress) or 30% (severe water stress) for 35 d [58]. The height of each plant was measured every 10 days. The plant aboveground biomass, leaf RWC, *Pn*, and chlorophyll content were measured after 35 days.

### 4.10. Physiological Analysis

The Hydrogen peroxide (H_2_O_2_) content detection kit (Solarbio, Beijing, China) and Malondialdehyde (MDA) assay kit (TBA method) (Jiancheng, Nanjing, China) were used to determine H_2_O_2_ and MDA content in the poplar leaves, respectively. The 1 g of fresh leaves were ground evenly in the mortar together with 1 mL frozen acetone and a small amount of quartz sand. The absorbance of the supernatant was measured at 410 nm after centrifugation for 10 min.

At 10:00 am on a sunny day, the *Pn*, *Gs*, and *Tr* in leaves of the WT, 35S::-GFP, and OE plants were determined by CIRAS-3 (PP Systems, Amesbury, MA, USA) photosynthesis apparatus.

Firstly, chlorophylls were extracted from the isolated leaves of the WT, 35S::-GFP, and OE plants with 80% acetone, and absorbance at 663 and 645 nm was measured using a NanoDrop 2000 instrument (Thermo Fisher Scientific, Waltham, MA, USA), and then the chlorophyll contents were calculated [58]. The above-ground parts of the harvested plants were killed by heating at 105 °C for 15 min, and dried to a constant weight at 65 °C to obtain the biomass.

### 4.11. Stomatal Opening Observation

In order to understand the change in stomatal opening of transgenic poplar under drought stress, the leaves of 2-month-old WT, 35S::GFP, and OE plants were separated, and the samples were fixed as described by Wang [58]. First, the leaves were perforated with a perforator at the same position on both sides of the main leaf vein. The leaves were immersed in a solution containing 0.01 M KCl, 0.5 M CaCl_2_, and 0.1 M MES-KOH under light for 2 h, and then 10 μM ABA was added. The pores were fixed after ABA treatment for 0, 1, and 2 h. The stomata were observed by a scanning electron microscope (FEI Quanta 200, Hillsboro, OR, USA). Each pore size datapoint included 3 biological replicates.

## 5. Conclusions

In this study, *PmbHLH58* was cloned from *P. massoniana*, an important forestry industry tree species used mainly for turpentine production and wood production in China. To study the role of *PmbHLH58* in plants under drought stress, the pSuper:: *PmbHLH58* vector was transformed into *Populus davidiana* × *P. bolleana*. The overexpression of *PmbHLH58* can improve drought tolerance by regulating stomatal opening and photo-cooperation, and may be involved in ABA-dependent stress response pathways. The regulatory relationship between PmERF71 and PmbHLH58 was found by the yeast double-hybrid technique, which can provide help for further research on drought tolerance pathways. The results of this study are not only helpful in understanding the drought tolerance mechanism of *PmbHLH58* in *P. massoniana*, but also the new poplar germplasm formed by the transgenic plants may provide breeding materials for the acquisition of new poplar varieties.

## Figures and Tables

**Figure 1 ijms-26-00277-f001:**
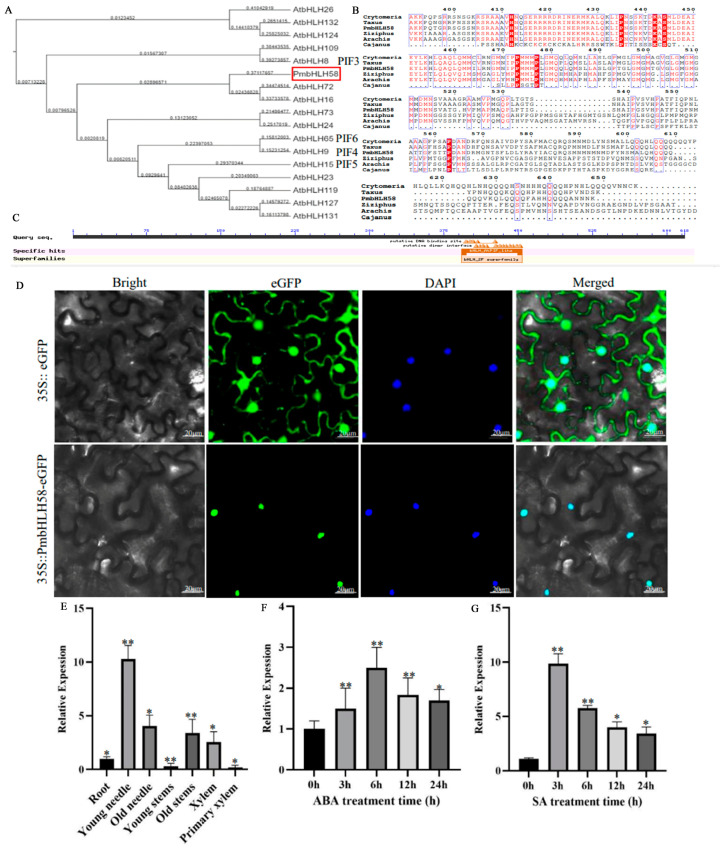
*PmbHLH58* of *P. massoniana*. (**A**) The phylogenetic tree of the *PmbHLH58* of *P. massoniana* and *A. thaliana*. The different surrounding letters represent different groups. (**B**) Multiple alignments of amino acid sequences of *PmbHLH58* and other plant bHLH proteins. (**C**) The bHLH superfamily domain of *PmbHLH58*. (**D**) The subcellular localization of the *PmbHLH58* protein. (**E**) The transcript levels of PmbHLH58 in various tissues. (**F**) *PmbHLH58* transcript levels under ABA stress. (**G**) *PmbHLH58* transcript levels under SA stress. The bars represent significant differences (* *p* < 0.05, ** *p* < 0.01).

**Figure 2 ijms-26-00277-f002:**
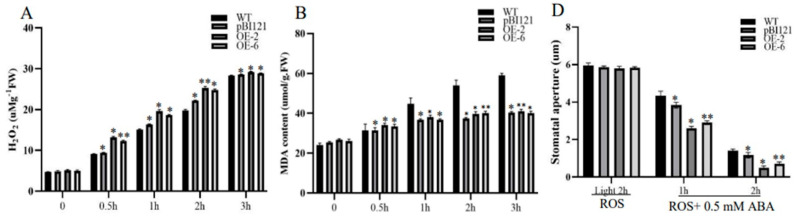
*PmbHLH58* promotes drought stress-induced stomatal closure via reactive oxygen species (ROS) accumulation. (**A**) H_2_O_2_ content analysis. (**B**) Stomatal apertures assay. (**C**) Stomatal activity. (**D**) Stomatal apertures assay. Asterisks represent the significance of differences (* *p* < 0.05, ** *p* < 0.01).

**Figure 3 ijms-26-00277-f003:**
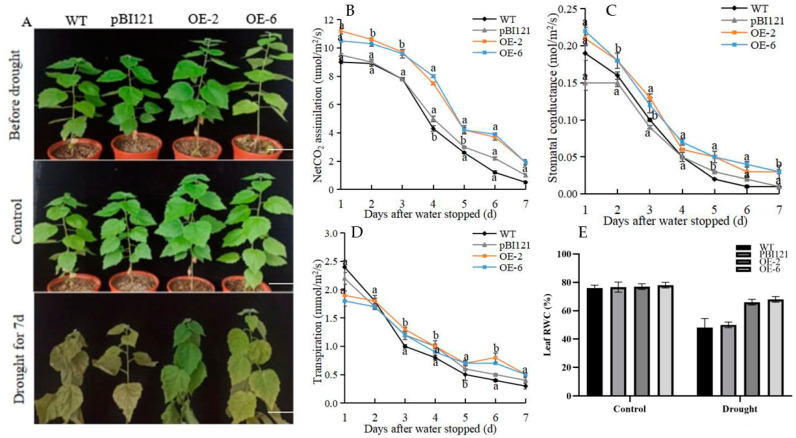
Overexpression of *PmbHLH58* enhanced drought tolerance under short-term drought conditions. (**A**) Phenotype observation of plants under short-term drought conditions. Scale bars = 10 cm. (**B**) Net photosynthetic rate. (**C**) Stomatal conductance. (**D**) Transpiration rate. Means with different letters are significantly different at *p* < 0.05. (**E**) Leaf RWC under non-stress and drought conditions. Asterisks denote significant differences: * *p* < 0.05.

**Figure 4 ijms-26-00277-f004:**
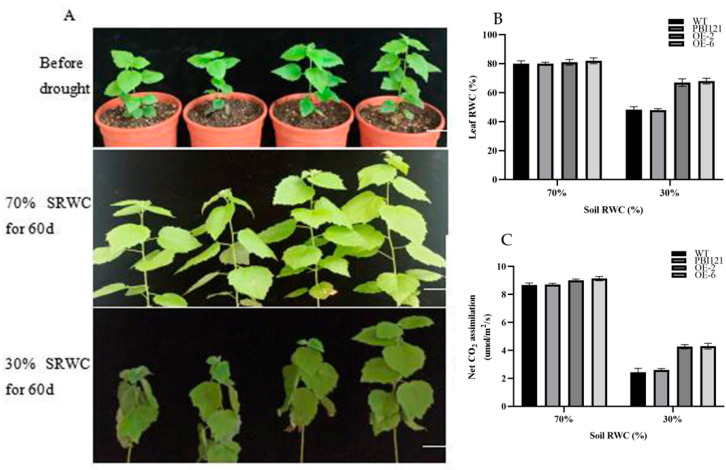
Overexpression of *PmbHLH58* enhanced drought tolerance under long-term drought conditions. (**A**) Phenotype observation of plants in long-term drought experiments. Scale bars = 10 cm. (**B**) Leaf RWC under different water conditions. (**C**) Net photosynthetic rate. Asterisks denote significant differences: * *p* < 0.05; ** *p* <0.01.

**Figure 5 ijms-26-00277-f005:**
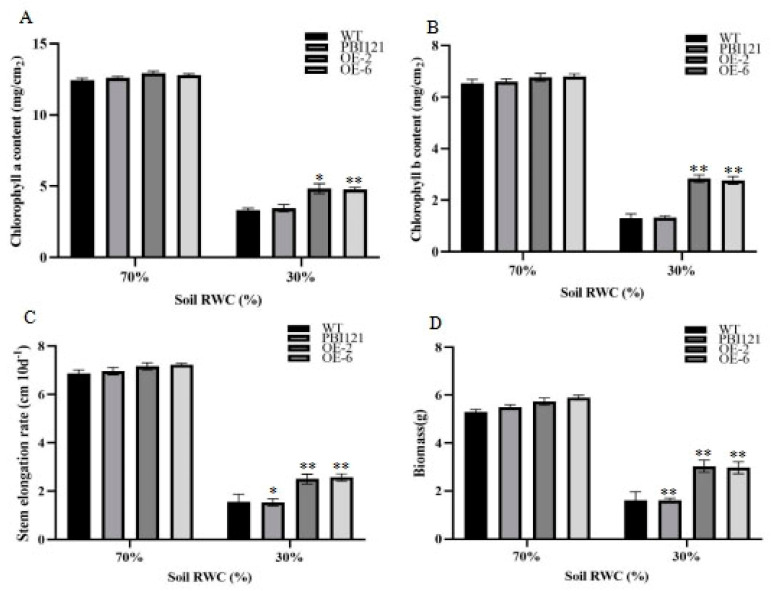
The physiological analysis of plants overexpressing *PmbHLH58* under long-term drought conditions. (**A**) The content of chlorophyll a. (**B**) The content of chlorophyll b. (**C**) Whole-plant biomass. (**D**) Stem elongation rate. Data are means ± SE (*n* = 6). Asterisks denote significant differences: * *p* < 0.05; ** *p* < 0.01.

**Figure 6 ijms-26-00277-f006:**
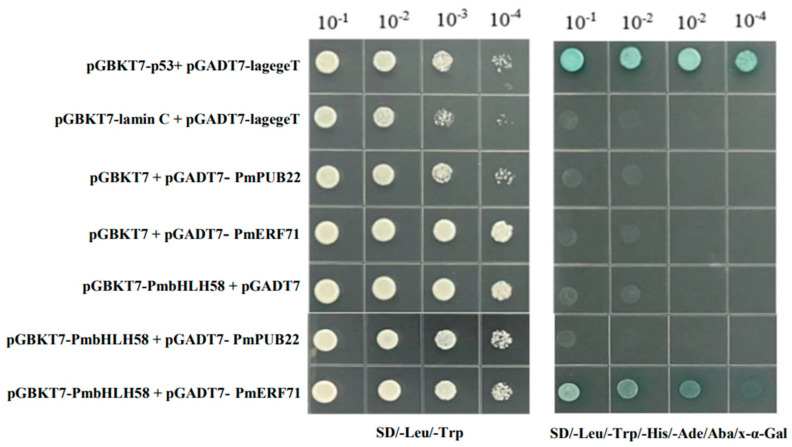
Point board verification of pGADT7-PmERF71 + pGBKT7-PmbHLH58 and pGADT7-PUB22 + pGBKT7-PmbHLH58.

**Figure 7 ijms-26-00277-f007:**
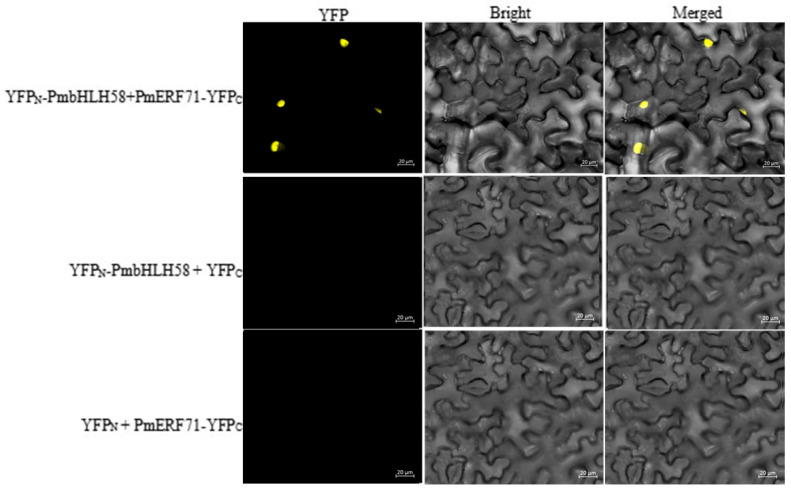
Bimolecular fluorescence complementation assays of the interaction between PmbHLH58 and PmERF71. The YFP signal was detected only in *N. benthamiana* leaves co-transformed with the YFP_N_-PmbHLH58 and PmERF71-YFP_C_ vectors. YFP_N_-PmbHLH58/empty YFP_C_ and empty YFP_N_/PmERF71-YFP_C_ were used as negative controls. Scale bars are 20 μm.

**Figure 8 ijms-26-00277-f008:**
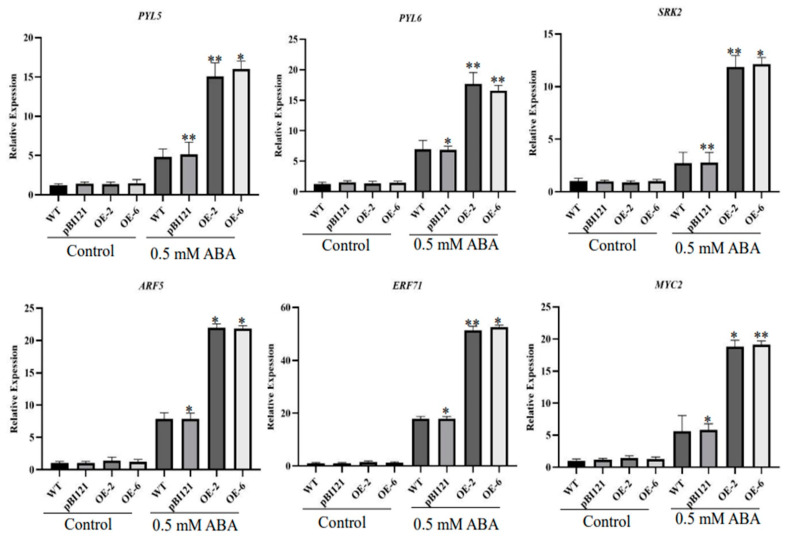
The expression of ABA-responsive genes under control and drought stress conditions. The significance of differences between the different lines was tested with the *t*-test. Asterisks represent significant differences between each OE line and the WT (* *p* < 0.05, ** *p* < 0.01).

**Figure 9 ijms-26-00277-f009:**
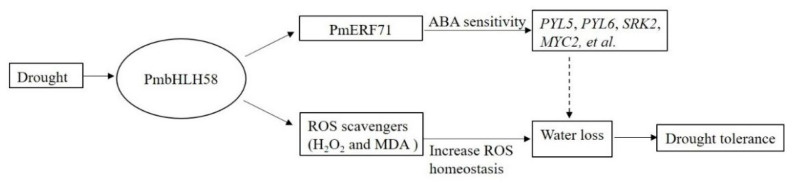
A proposed model explaining the function of *PmbHLH58* in response to drought stress.

**Table 1 ijms-26-00277-t001:** List of proteins interacting with PmbHLH58 by yeast two-hybrid screening.

Protein ID	Species	Protein Name
XM_035057714.1	*P**opulus* *alba*	disease resistance protein At4g27220
XM_035071162.1	*P* *. alba*	proline-rich protein-like (LOC118058459)
XM_035055231.1	*P. alba*	D-3-phosphoglycerate dehydrogenase 1
XM_011007353.1	*P. euphratica*	extra-large guanine nucleotide-binding protein
XM_035036041.1	*P. alba*	UDP-N-acetylglucosamine transporter
XM_035033747.1	*P. alba*	acidic leucine-rich nuclear phosphoprotein
XM_035048879.1	*P. alba*	ethylene-responsive transcription factor ERF71
XM_035075021.1	*P. alba*	glu S.griseus protease
XM_035072978.1	*P. alba*	40S ribosomal protein S23
XM_035035937.1	*P. alba*	probable NAD(P)H dehydrogenase
XM_035047444.1	*P. alba*	actin-depolymerizing factor
XM_035067037.1	*P. alba*	glycine-rich RNA-binding protein 2
XM_035061321.1	*P. alba*	E3 ubiquitin-protein ligase PUB22
XM_035051584.1	*P. alba*	PLAT domain-containing protein
XM_035036084.1	*P. alba*	annexin D2
XM_011045048.1	*Populus*	euphratica cytochrome c oxidase subunit 2
XM_035043888.1	*P. alba*	ankyrin repeat-containing protein At5g02620
XM_035043888.1	*P. alba*	proline-rich receptor-like protein kinase PERK8
XM_035032624.1	*P. alba*	superoxide dismutase [Mn]
XM_035053814.1	*P. alba*	NADH dehydrogenase [ubiquinone] 1 alpha subcomplex assembly factor 3
XM_035038738.1	*P. alba*	osmotin-like protein OSM34
XM_035073701	*P. alba*	60S acidic ribosomal protein P2B
XM_035032988.1	*P. alba*	mini zinc finger protein 2
XM_011025726.1	*P. euphratica*	SNF1-related protein kinase regulatory subunit
XM_035074523.1	*P. alba*	TrmH family tRNA/rRNA methyltransferase
XM_035051580.1	*P. alba*	acyl-coenzyme A oxidase 2
XM_002316029.3	*P. trichocarpa*	protein translation factor SUI1 homolog

## Data Availability

Data are contained within the article and Appendix A.

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
