# Peer review of "PmbHLH58 from Pinus massoniana Improves Drought Tolerance by Reducing Stomatal Aperture and Inducing ABA Receptor Family Genes in Transgenic Poplar Plants"

_ijms, 2024, doi:10.3390/ijms26010277_

Round 1
Reviewer 1 Report
Comments and Suggestions for Authors
The paper makes a very complete functional description of PmbHLH58 a gene which plays a significant role in improving drought tolerance in plants, and demonstrates its functionality in transgenic poplar plants. Authors convincingly show that the gene is involved in key physiological processes such as regulation of stomatal aperture, activation of ABA signaling and improved drought tolerance, and finally the show that is a potential candidate for biotechnological improvement of trees.
I only have some minor comments:
1. The study mentions that reducing stomatal conductance can prevent excessive water loss but this may also negatively impact photosynthesis and concomitantly biomass accumulation. All the experiments have been performed under drought stress and in short term, but there is a possiblity that under standard growing conditions and in long term, the gene overexpression induce a yield penalty or plants exhibit lesser growth. Have the authors cosider this possibility? Please comment this in the discussion.
2. The document uses terms like "drought tolerance" and "drought resistance" interchangeably. I advice to use the same tarm, as they are not fully equivalent.
3. In figure 1 the legend is too concise and not really informative. Plese include the following information: Pannels A and B: which software was used and which parameters. Pannel d: where was the gfp-protein expressed? Nicotiana? it was transient expression with agorbacterium? Pannels E, F and G how many replicas per bar (please include the n number as in other figures). Which was the statistical analysis performed? Please include this information in all the figure legends.
Table 1 legend: "screening"
Reviewer 2 Report
Comments and Suggestions for Authors
Comments
Dear Author,
It is my pleasure to review the manuscript entitled “PmbHLH58 from Pinus massoniana Improves Drought Tolerance by Reducing Stomatal Aperture and Inducing ABA Receptor Family Genes in Transgenic Poplar Plants” a research article submitted to MDPI Journal, IJMS. Authors of this manuscript cloned and characterized PmbHLH58 gene in Pinus massoniana. They used a series of bioinformatic and lab experiments to identify function of this gene in the transgenic plant including subcellular localization, overexpression, yeast two-hybrid and TEM. They also identified interaction of PmERF71 with PmbHLH58. Finally concluded its function in drought tolerance. The overall experiments, they performed, are well and the results are very convincing and important for cultivation. Thus, the presented results take up an important topic consistent with the profile of the Journal.
Most suggestions I made in the original pdf. Please check carefully which might improve the manuscript to make important to the wider readers.
-English must be improved significantly for clarity of research.
Abstract:
-Genes and scientific name should be italic, throughout the text.
Introduction:
- With more constructive rationale of the study, elaborate clearly in introductory, why this research is necessary and identify gaps of the previously published research.
-What about PmbHLH58 gene in introductory? Is it new? No work done in P. massoniana? What about other plants? You need to introduce with proper references.
Results:
-Line 103; not well-structured sentence.
-Cloning and gene identification should be the first result
L104-107; How plant hormone signal, circadian rhythm linked drought??? If it is so, ypou need to discuss in introduction with proper references.
-Needs more writing about phylogenetic tree analysis
L132-136; Not clear information.
-Some figs, quality not good. Improve resolution. 1C not legible
-Fig. 1; Title of fig. is not enough. Write full statement.
-L147-152; Incomplete information. PCR using which sample? What was primers?
Material
-Some sentences are very long, many repeated wording in the same sentence or even subsequent sentence. That distract continuous flow of the reading.
-PCR components are not listed correctly. Use of µL does not give exact amount. Need to use ng. Check throughout the text.
Percent match: 40%. It should be below 20%

The English could be improved to more clearly express the research.
Reviewer 3 Report
Comments and Suggestions for Authors
Receptor Family Genes in Transgenic Poplar Plants.
This type of knowledge is helps to these findings provide new insights into transcriptional regulation mechanisms related to drought stress and will promote the progress of genetic improvement and plantation development of P.massonsiana.
Droughts are estimated to affect 55 million people worldwide each year, posing the most serious threat to livestock and crops in almost every part of the world. Rising temperatures due to climate change are already making arid regions drier.
As the authors emphasize “…Plants are subjected to biological and abiotic stresses during their growth, drought is one of the most important abiotic factors that hinder plant growth. As the global climate changes, temperatures and droughts will continue to increase…”.
The aim of this study was to expand the regulatory role of bHLH in drought resistance. The authors selected high expression bHLH gene from transcriptome data under drought stress for cloning.
The paper deals with important issues:
· Expression Pattern Analysis of PmbHLH58;
· Heterologous Expression of PmbHLH58 in Populus davidiana × P.bollean;
· PmbHLH58 promotes ABA-induced stomatal closure;
· Under Short-term and Long-term Drought Stress OE Lines Showed Improved Drought Tolerance;
· The PmbHLH58 Protein Interacts with PmERF71;
· Analysis of Key Enzyme Genes in ABA Hormone Signaling Pathway.
The article presents the research material, methods used in a given research work in a clear and transparent way, and discusses it thoroughly. The site description and experimental design are easy to understand. This experiment seems to be well planned and clearly explained in the description of the methodology. The discussion is written correctly and to the point. Clearly reflected the reference to the presented results.
Abstract clearly informs about the experiment performed. The methodology is clear and described concisely. Introduction section is good and is also written in an concise and clear manner. The literature is well-chosen, current and the conclusions clearly refer to the conducted research. Research confirmed that overexpression of PmbHLH58 can improve drought resistance by regulating stomatal opening and photo-cooperation, and may be involved in ABA-dependent stress response pathways.
As the authors note in the conclusions this study are not only helpful to understand the drought resistance mechanism of PmbHLH58 in P. massoniana, but also the new poplar germplasm formed by transgenic may provide breeding materials for the acquisition of new poplar varieties.
There are some minor weaknesses in article, therefore I recommend minor revision of the paper.
Minor issues to be corrected:
· Introduction
Line 38-40 – It is worth adding a citation of literature or the latest reports regarding „changes, temperatures and drought”.
Line 78 – It is worth checking, showing or adding the latest literature on plants overexpressing resistance to abiotic stress.
Line 94 – Word “a few” – unnecessary “Physiological indexes and photosynthetic characteristics were analyzed 94 during the drought treatment to determine if the transgenic plants had improved drought resistance compared to the wild type plants.
· Results
Figure 1 - It may be worth separating the drawings. Describe each drawing as a separate figure. The drawings will be larger and clear. Figure ABC can be collectively described as figure 1. Similarly in the other cases.
If the program allows you to enlarge the font in figure 1A it is worth applying more readable font.
· Discussion
Line 290 - The beginning of the discussion, it is worth providing literature data on the topic the drought seriously affects the growth and development of plants.
Line 310-311 – These results indicate that PmbHLH58 may be involved in the response process of P. massoniana to drought stress. These results indicate that PmbHLH58 may be involved in the response process of P. massoniana to drought stress. Repeated sentences
Line 314 – It is worth quoting other authors here, maybe you will be able to find newer literature. The sentence is quite crucial due to the subject of the work.
Figure 9 – describe as source data or your own work
· Materials and Methods
Line 403-404 – sentences formulated incorrectly.
Line 403-404 – according to what procedure/methodology?
Line 442 – explain why for 7 days?
The summary is sufficient and clearly reflects the obtained research results.
In summary, the paper is worth publishing in the Journal after considering the above comments.
Round 2
Reviewer 2 Report
Comments and Suggestions for Authors
Even though, there is no specific response for the comments, however, article has been somewhat improved.
Need further improvement. English correction needed
Line 25; "it" should be "TF"
No change in line 123, 140. Figs. were not improved. Line 247-258; nothing was changed. Nothing was changed in line 380-388. Line 391-395; nothing is improved. Fig; 1.; no change from previous comment
